# *Fruit-In-Sight*: A deep learning-based framework for secondary metabolite class prediction using fruit and leaf images

**Neeraja M. Krishnan**[1]*, **Saroj Kumar**[1], **Binay Panda**[1,2]*

**1** School of Biotechnology, Jawaharlal Nehru University, New Delhi, India, **2** Special Centre for Systems Medicine, Jawaharlal Nehru University, New Delhi, India

\* neeraja@jnu.ac.in (NMK); binaypanda@jnu.ac.in (BP)

## Abstract

Fruits produce a wide variety of secondary metabolites of great economic value. Analytical measurement of the metabolites is tedious, time-consuming, and expensive. Additionally, metabolite concentrations vary greatly from tree to tree, making it difficult to choose trees for fruit collection. The current study tested whether deep learning-based models can be developed using fruit and leaf images alone to predict a metabolite's concentration class (*high* or *low*). We collected fruits and leaves ($n = 1045$) from neem trees grown in the wild across *0.6* million sq km, imaged them, and measured concentration of five metabolites (azadirachtin, deacetyl-salannin, salannin, nimbin and nimbolide) using high-performance liquid chromatography. We used the data to train deep learning models for metabolite class prediction. The best model out of the seven tested (*YOLOv5*, *GoogLeNet*, *InceptionNet*, *EfficientNet_B0*, *Resnext_50*, *Resnet18*, and *SqueezeNet*) provided a validation *F1* score of 0.93 and a test *F1* score of 0.88. The sensitivity and specificity of the fruit model alone in the test set were 83.52 ± 6.19 and 82.35 ± 5.96, and 79.40 ± 8.50 and 85.64 ± 6.21, for the *low* and the *high* classes, respectively. The sensitivity was further boosted to 92.67± 5.25 for the *low* class and 88.11 ± 9.17 for the *high* class, and the specificity to 100% for both classes, using a multi-analyte framework. We incorporated the multi-analyte model in an Android mobile App *Fruit-In-Sight* that uses fruit and leaf images to decide whether to 'pick' or 'not pick' the fruits from a specific tree based on the metabolite concentration class. Our study provides evidence that images of fruits and leaves alone can predict the concentration class of a secondary metabolite without using expensive laboratory equipment and cumbersome analytical procedures, thus simplifying the process of choosing the right tree for fruit collection.

## Introduction

Secondary metabolites, especially those derived from microbial and plant sources, have a range of industrial applications, such as drugs, fragrances, dyes, pigments, pesticides, and food additives. Analytical (chemical, biochemical, immunological, or imaging-based) methods are routinely used in laboratories for measuring secondary metabolites. Some techniques, like

**Data Availability Statement:** All data are freely available at https://www.kaggle.com/datasets/binaypandalabmember/plos-one-data/

**Funding:** Research reported in this manuscript is funded by an extramural grant from the

Department of Biotechnology, Government of India to BP (BT/PR36744/BID/7/944/2020). The funders had no role in study design, data collection and analysis, decision to publish, or preparation of the manuscript.

**Competing interests:** The authors have declared that no competing interests exist.

imaging, although simple to use, could be more precise. Accurate and precise methods, like high-performance liquid chromatography (HPLC), require extensive sample handling, long preparation time, expensive equipment, and specialized skills. Often, a quick and rough categorization of the metabolite concentration into *high* or *low* class is enough to choose suitable fruits for industrial purposes.

Artificial intelligence-based methods, with their proven utility in various aspects of plant science, are becoming integral to plant phenomics research [1, 2]. Deep learning methods outperform conventional machine learning methods of image classification like Support Vector Machines and Random Forest classifiers that rely on extraction of features such as principal components identification [3]. In the present study, we have used images of fruits and leaves from neem (*Azadirachta indica*) trees to test whether deep learning-based methods can predict the concentration class (*high* or *low*) of metabolites that give the fruit its characteristic value. Neem fruits are a valuable source of secondary metabolites, including the potent anti-feedant azadirachtin, a widely used alternative to chemical pesticides [4].

In addition to the cumbersome laboratory processes and cost, a factor that makes metabolite extraction less effective is a variation in the azadirachtin concentration from tree to tree, even within a small area [5, 6]. As the commercial extraction of azadirachtin uses fruits from multiple trees, metabolite yield varies significantly from batch to batch. Therefore, a quick and inexpensive method to categorize metabolite concentration at source will substantially help select trees that bear high concentrations of metabolites in their fruits and improve yield.

## Materials and methods

### Images used in the study, secondary metabolite concentration measurement and preparation of images for deep learning models

We collected fruit and leaf images (*n = 1045*) from trees spread across *0.6* million $Km^2$ to build a training set of images. We measured the concentrations of five metabolites - azadirachtin (*A*), deacetyl-salannin (*D*), nimbolide (*E*), nimbin (*N*) and salannin (*S*) after pooling five fruits from the same tree using reverse phase columns on the high-performance liquid chromatography (HPLC) instrument, with respective analytical standards (EID Parry, India).

The images were divided into *high* and *low* classes for each metabolite using the respective mean concentrations as the thresholds. We drew bounding boxes around the fruit contours using Makesense.ai (https://www.makesense.ai/)) to get the object's annotation coordinates. Further, we augmented the training set with ~10% background images chosen randomly from openly accessible and other images (128 images from *coco-128*; https://www.kaggle.com/datasets/ultralytics/coco128 and images of 54 flowers, 50 leaves, and 50 fruits randomly sourced from *Google Images*). While we treated the single-analyte framework for fruit *A* class prediction as a regular object detection-cum-classification problem, the multi-analyte classification framework involved adding a combinatorial approach of boosting fruit *A* class predictions using the predictions from the *D*, *E*, *N*, and *S* fruit models based on the fruit images and *A*, *D*, *E*, *N* and *S* leaf models based on the leaf images. We could do this by measuring all five metabolites across the same fruits and imaging the fruits and leaves from the same tree. Fig 1 provides the flow for the single- and multi-analyte deep learning frameworks.

### Optimization of training parameters and predictive models

We tested multiple variants of the PyTorch-based '*You Only Look Once' YOLO* (v5) [7] framework's medium (*m*) architecture for object detection and image classification to predict the metabolite concentration class of detected fruit and leaf objects. The *YOLOv5* framework for

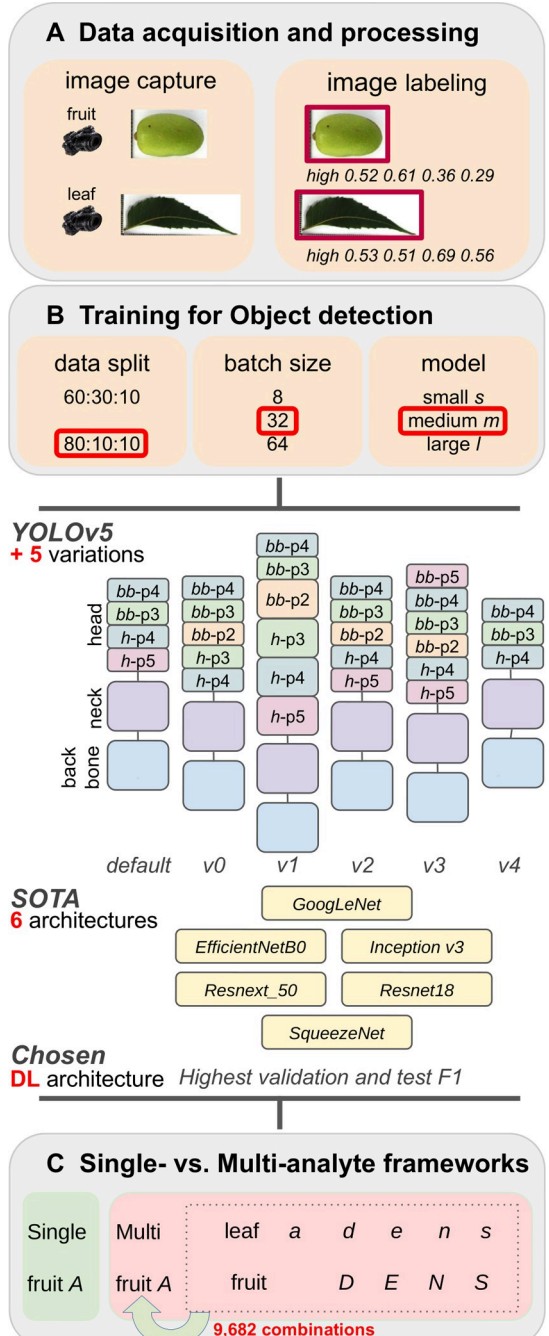

**Fig 1. Overview of the single- and multi-analyte deep learning framework phases for class prediction in neem fruit and leaf images.**

object detection is known for its high runtime speeds without loss of accuracy. Other than the default *YOLOv5m* architecture, we tested five variants: *v0* (the *YOLO* architecture adapted to detect small objects (http://cs230.stanford.edu/projects_fall_2021/reports/103120671.pdf)), *v1* (modified head by tweaking *v0* and increasing the number of C3 layers in the p2 block from 1 to 3, in the p3 block from 3 to 5, in the p4 block from 3 to 5 and added a p5 block), *v2* (added an extra p2 block to head), *v3* (added 1 extra p2 and p5 blocks each to the head), and *v4*

(deleted p5). We also tested six state-of-the-art architectures under image classification frameworks such as *GoogLeNet* [8], *Inception v3* [9], *EfficientNet_B0* [10], *Resnext_50* [11], *Resnet18* [12] and *SqueezeNet* [13] using default parameters. The image classification frameworks used cropped images after bounding box detection with the best model obtained from *YOLOv5*.

We exported the bounding box labels in the five-coordinate *YOLOv5* format. We used an 80:10:10 split ratio of the dataset into train, validation and test subsets, batch size of 64, and the medium variant of the *YOLOv5* model, characterized by a neural network with 21.2 million parameters. We monitored and recorded the performance metrics, Precision (*P*), Recall (*R*), mAP:0.5 (*M1*), mAP:0.5:0.95 (*M2*) and F1-score ($F1 = 2 \times P \times R (P + R)$) and based the selection of model on maximizing *F1*. We also monitored the loss curves (object, box and class) for training and validation to prevent over-fitting. We used Weights & Biases (http://wandb.com/; [14]) integration with *YOLOv5* to track, log and visualize all the *YOLOv5* runs. We trained all runs for 300–2000 epochs with the default hyper-parameter set. After identifying the epoch at which over-fitting occurs based on the validation loss curve crossing over the training one, we retrained until that epoch to obtain the best models for all deep learning frameworks.

## Single- and multi-analyte frameworks for class prediction

We first individually predicted the metabolite concentration classes of images from the unused test set using each of the ten models from the single-analyte framework. We further identified patterns by combining predictions from the *A*, *D*, *E*, *N* and *S* models derived by training using fruit and leaf images ($A_f$, $D_f$, $E_f$, $N_f$, $S_f$, $A_l$, $D_l$, $E_l$, $N_l$ and $S_l$). In doing so, we performed various combinations of the predictions from the ten models, ranging from single models, pairs of models, . . ., up to a combination of all ten models. We used these combinatorial patterns to boost the prediction accuracy of fruit *A*. We termed this the 'multi-analyte' framework for class prediction from multiple metabolites and image types.

## Cross-validation

We performed 10-fold cross-validation on the neem image dataset by making ten random data splits into sets of 80:10:10 for training, validation, and testing subsets. For each random split, we trained and tested the prediction efficiency of the best models using the test set images. We obtained the prediction error for each split by comparing the predicted class to the actual class, and averaged these across all ten splits to obtain the cross-validation error in prediction.

The boosted fruit *A* prediction accuracy was estimated for the *low* and *high* classes for all tuple combinations of the ten models: pairs ($A_fD_f$, $A_fE_f$, $A_fN_f$, . . ., $N_lS_l$), 3-tuples ($A_fD_fE_f$, $A_fD_fN_f$, $A_fD_fS_f$, . . ., $E_lN_lS_l$), 4-tuples ($A_fD_fE_fN_f$, $A_fD_fE_fS_f$, $A_fD_fN_fS_f$, . . ., $D_lE_lN_lS_l$), 5-tuples ($A_fD_fE_fN_fS_f$, $A_fD_fE_fN_fA_l$, $A_fD_fE_fN_fD_l$, . . ., $A_lD_lE_lN_lS_l$), 6-tuples ($A_fD_fE_fN_fS_fA_l$, $A_fD_fE_fN_fS_fD_l$, $A_fD_fE_fN_fS_fE_l$, . . ., $S_fA_lD_lE_lN_lS_l$), 7-tuples ($A_fD_fE_fN_fS_fA_lD_l$, $A_fD_fE_fN_fS_fA_lE_l$, $A_fD_fE_fN_fS_fA_lN_l$, . . ., $N_fS_fA_lD_lE_lN_lS_l$), 8-tuples ($A_fD_fE_fN_fS_fA_lD_lE_l$, $A_fD_fE_fN_fS_fA_lD_lN_l$, $A_fD_fE_fN_fS_fA_lD_lS_l$, . . ., $E_fN_fS_fA_lD_lE_lN_lS_l$), 9-tuples ($A_fD_fE_fN_fS_fA_lD_lE_lN_l$, $A_fD_fE_fN_fS_fA_lD_lE_lS_l$, $A_fD_fE_fN_fS_fA_lD_lN_lS_l$, . . ., $D_fE_fN_fS_fA_lD_lE_lN_lS_l$) and 10-tuple ($A_fD_fE_fN_fS_fA_lD_lE_lN_lS_l$) where each metabolite was classified as *low* or *high* respectively. Then, the combinations that predicted the $A_f$ class with complete specificity were identified. Such combinations would only predict *low* $A_f$ or *high* $A_f$ class but not both, and could be predictive of *low* $A_f$ or *high* $A_f$ class across test dataset splits with varying sensitivity, ranging from 1 to all 10.

## The Android App *Fruit-In-Sight*

We developed *Fruit-In-Sight* using the latest UI-based *Flutter* framework in its default *Dart* language (https://resocoder.com/2019/06/01/flutter-localization-the-easy-way-internationalization-with-json/). The reactive UI framework provided by *Flutter* not only allows user interface changes to be triggered by state change as in other reactive frameworks like *React*/*Vue* but also for the application to be created and run natively, unlike *React* which uses *HTML*.

The *YOLOv5* models were converted to *PyTorch mobile* to be compatible with *Flutter*. The pictures taken in the mobile app were pre- and post-processed such that the results obtained on the mobile platform matched the results obtained using the *YOLOv5* detect module for class prediction in test set images. The pre-processing involved resizing the image by aspect-fitting the image to 640 X 640 to match the dimension of the training images as per the *YOLOv5* utils.Augmentations.letterbox function. Following are the pre-processing steps:

1. Find the scaling factor to reduce the size of the image

2. Create a new blank image

3. Resize the original image using the scaling factor to match the model shape

4. Calculate the remaining space according to the target width and target height

5. Fill this with the color that *YOLOv5* uses for training, in our case (114,114,114)

6. Paste the resized image to get the image shape required by the model

7. Transpose the image

8. Convert the image from BGR to RGB

9. Convert the image color data range from 0 to 255 to 0 to 1.

*YOLOv5* detects many objects post-training with many possible rectangles in a single pass inference with confidence value for each rectangle. *YOLOv5* performs post-processing using Non-Max Suppression (*NMS*) to merge the overlapping rectangles that may have the same object and class, and provides results with confidence rate above 0.25. In the absence of similar post-processing, the shape of the output tensor is [16, 32256, 6]. To match results with that from the *YOLOv5* detect module, we applied *NMS* to 1, 2 or 3 rows and considered only the top result. We implemented the *NMS* logic in *Dart* due to lacking suitable *Flutter* library.

*Fruit-In-Sight* requires user authentication through a mobile number and a one-time password (OTP), done using *Firebase* authentication. We use *Azure* storage containers to store the model assets and *Firebase* to store the inferences run by users, along with images used for prediction and results shown to the user. *Firebase* provides *Flutter* SDK, which allows us to easily use *Firebase* services without having to write code to access webservice/API through *HTTP* or *RPC* protocols directly (https://firebase.google.com/docs/flutter). We used the Android Play Store to distribute *Fruit-In-Sight* (https://play.google.com/store/apps/details?id=org.binaypandalab.fruitinsight). Multilingual support is implemented through OS and platform-level features to take advantage of the Language and Locals features of the Android OS. The conceptual workflow of *Fruit-In-Sight* is shown in Fig 2.

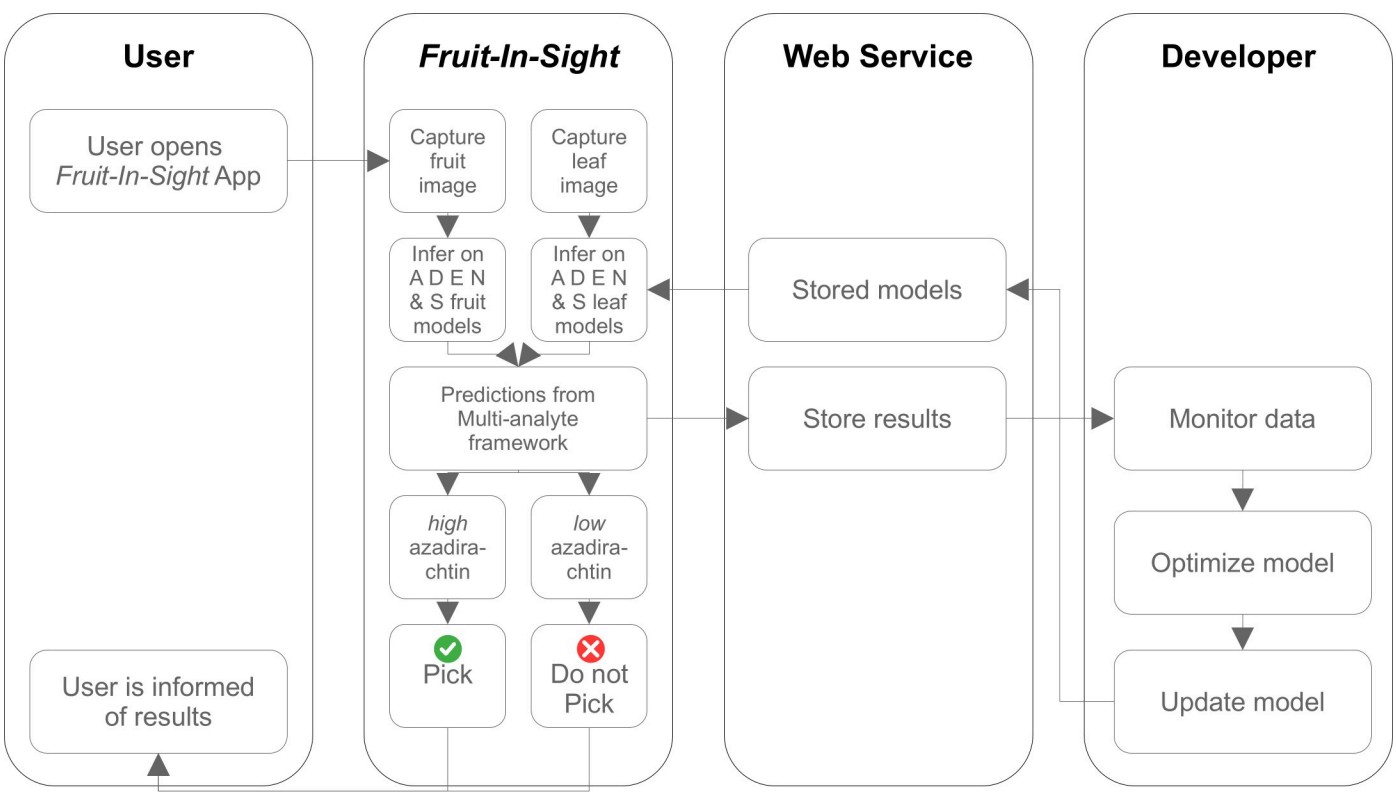

**Fig 2. Conceptual workflow of the mobile application, *Fruit-In-Sight*.**

## Results

### Metabolite concentration ranges and class formation

For neem fruit metabolites *A*, *D*, *E*, *N* and *S*, the concentrations (units) ranged from 0.181 to 1.003, 0.007 to 0.691, 0.004 to 0.252, 0.009 to 0.501, and 0.056 to 1.42, respectively, with 0.563, 0.112, 0.045, 0.163, and 0.511, as respective means. Fig 3 shows the distributions for these metabolite concentrations. We used the mean values as thresholds for concentration values below which the images are labeled *low* and above, which are labeled *high*, for the respective metabolites. The numbers of fruit and leaf images in the *low* and *high* classes across the train, validation and test splits of the dataset are specified in Table 1.

### Validation and test set performance metrics for the fruit *A* model

We estimated five performance metrics for the neem fruit *A* model using the various deep learning frameworks on the validation and test sets (see Methods). Of these, we prioritized the *F1* score, a harmonic mean of precision and recall, to compare the performance of the various deep learning frameworks. For the combined classes and the *high* class, the *YOLOv5m* default model yielded the highest *F1* score of 0.93 in the validation set, with the training data augmented with the background images meant as negative control (Table 2). This was reflected in the two mean average precision (*mAP*) scores: *M1* and *M2*. The performance of this model, as measured by the *F1* score and the *M1*, was the best for both classes combined and the *low* class even in the test set (Table 2). The *v3* variant of the *YOLOv5m model* performed the worst of all in terms of *F1*, *M1* and *M2*. In general, the variants *v0-v4* of the *YOLOv5* model did not yield

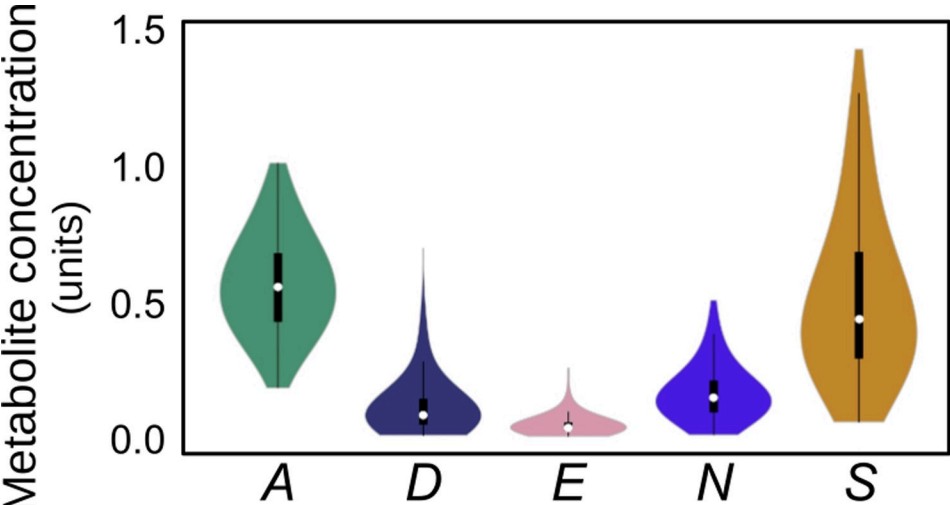

**Fig 3. Distribution of metabolite concentration values.** Concentrations of *A*, *D*, *E*, *N* and *S* neem metabolites are plotted as box plots with the data points overlaid as a bee swarm scatter. Center lines show the medians; box limits indicate the 25[th] and 75[th] percentiles; whiskers extend 1.5 times the inter-quartile range from the 25[th] and 75[th] percentiles; dots represent outliers; crosses represent sample means; data points are plotted as open circles. White circles show the medians; box limits indicate the 25[th] and 75[th] percentiles as determined by *R* software; whiskers extend 1.5 times the interquartile range from the 25[th] and 75[th] percentiles; polygons represent density estimates of data and extend to extreme values.

better results than the default version of this model, both with and without augmentation by negative control images (Table 2).

## Validation and test set performance metrics for other models

We further trained the fruit *D*, *E*, *N* and *S* models and leaf *A*, *D*, *E*, *N* and *S* models using the *YOLOv5m* architecture while including background images during training. We obtained corresponding validation and test set metrics (Table 3). The fruit *S* model resulted in the best overall performance in all the five validation and test set metrics, closely followed by the fruit *D* model (Table 3). The leaf *E* model resulted in the worst performance overall for validation and test sets.

## Ten-fold cross validation

We performed ten random shuffles of the data to avoid training bias in a different training, validation and test split. The sensitivity and specificity of the neem fruit *A* model alone to predict *low* and *high* classes of fruit *A* was estimated for each split. The sensitivity of prediction across ten random shuffles of the test set was 83.52 ± 6.19 and 82.35 ± 5.96 for *low* and *high* classes, respectively. The specificity of prediction of the same was 79.40 ± 8.50 and 85.64 ± 6.21, respectively.

## Boosting the performance of the fruit *A* model using multi-analyte framework

Predictions from the nine other models, namely, fruit *D*, *E*, *N* and *S* and leaf *A*, *D*, *E*, *N* and *S* models, were used to boost the predictions from the fruit *A* model. Patterns in the form of multi-analyte classification tuples that were predictive of *low* or *high* fruit *A* classes with 100% specificity, were identified and prioritized by their presence in as many dataset shuffles as

**Table 1. Number of neem fruit and leaf images within *high* (*h*) and *low* (*l*) classes of *A*, *D*, *E*, *N* and *S* metabolites among the training, validation and test datasets.**

| Plant analyte | Plant part | Class | Training | Validation | Test |
|---|---|---|---|---|---|
| *Azadirachtin (A)* | fruit | h | 360 | 45 | 50 |
| | | l | 465 | 50 | 75 |
| | leaf | h | 358 | 45 | 50 |
| | | l | 465 | 50 | 75 |
| *Deacetyl salannin (D)* | fruit | h | 420 | 45 | 55 |
| | | l | 405 | 50 | 70 |
| | leaf | h | 419 | 45 | 55 |
| | | l | 404 | 50 | 70 |
| *Nimbolide (E)* | fruit | h | 410 | 55 | 75 |
| | | l | 415 | 40 | 50 |
| | leaf | h | 408 | 55 | 75 |
| | | l | 415 | 40 | 50 |
| *Nimbin (N)* | fruit | h | 440 | 40 | 45 |
| | | l | 385 | 55 | 80 |
| | leaf | h | 440 | 40 | 45 |
| | | l | 383 | 55 | 80 |
| *Salannin (S)* | fruit | h | 415 | 50 | 55 |
| | | l | 410 | 45 | 70 |
| | leaf | h | 414 | 50 | 55 |
| | | l | 409 | 45 | 70 |

possible. Fig 4 depicts the average sensitivity of such predictive patterns as a function of the minimum number of dataset shuffles in which these patterns were predictive in the same direction. The numbers of predictive patterns ranged from 212, 637, 1400, 2462, 3801, 5512 and 7300 for the *low* fruit *A* class, and 32, 168, 453, 989, 1842, 3058 and 4503 for the *high* fruit *A* class for presence in 10, 9, 8, 7, 6, 5 and 4, dataset shuffles, respectively. The error bars represent the standard deviation in the sensitivity across the shuffles. The maximum sensitivities for *low* and *high* fruit *A* class prediction with complete specificity are 91.89% and 86.35% across the ten cross-validation test dataset shuffles, with standard deviations of 6.11% and 10.91%, respectively. The corresponding 10-fold cross-validation errors are 0.07 and 0.11 in the *low* and *high* fruit A classes, respectively. The prediction combinations under the multi-analyte framework that can be used for *low* and *high* fruit *A* class prediction are provided in S1 Table.

### *Fruit-In-Sight* Android mobile App

Using *Fruit-In-Sight* (https://play.google.com/store/apps/details?id=org.binaypandalab.fruitinsight) requires signing up using a mobile number and a One-Time-Password-based authentication. Once logged in, the user remains authenticated for further sessions until they log out. *Fruit-In-Sight* has two modules, one for neem metabolite prediction (this study) and the second for other fruits (we plan to include multiple fruits' imaging-based applications in the same App. The first part of the series is to determine the sweetness of a citrus fruit, kinnow. We shall cover the citrus fruit sweetness part of *Fruit-In-Sight* in a separate manuscript). *Fruit-In-Sight*'s neem module works serially on fruit and leaf images and makes a combined inference using the multi-analyte framework in the back-end. The suitability of the neem fruit is relayed to the user in the form of a 'Pick' or 'Do not Pick'.

**Table 2. Validation and test set metrics of the best neem fruit A models under the object detection using YOLOv5 medium variants and image classification on detected object categories.** The object detection category included the default YOLOv5m architecture and its five variations (v0, v1, v2, v3 and v4; see Methods), while the second category included six state-of-the-art image classification architectures. We studied the effect of adding random background images as negative control. The best models were estimated by retraining until epoch Ep when over-fitting was observed. Performance metrics included precision (P), Recall (R), F1 score (F1), mAP@0.5 (M1) and mAP@[.5,95] (M2), for both classes combined (a), as well as individually for the low (l) and high (h) classes.

Validation set metrics (object detection rows):

| Category | Architecture | Neg. Ctrl. | Ep. | P a | P l | P h | R a | R l | R h | F1 a | F1 l | F1 h | M1 a | M1 l | M1 h | M2 a | M2 l | M2 h |
|---|---|---|---|---|---|---|---|---|---|---|---|---|---|---|---|---|---|---|
| Object detection using YOLOv5 (medium) | default | No | 300 | 0.90 | 0.87 | 0.93 | 0.92 | 0.96 | 0.87 | 0.91 | 0.91 | 0.90 | 0.96 | 0.95 | 0.96 | 0.89 | 0.90 | 0.89 |
| | v0 | | 200 | 0.50 | 0.54 | 0.46 | 1.00 | 1.00 | 1.00 | 0.67 | 0.70 | 0.63 | 0.63 | 0.66 | 0.61 | 0.60 | 0.62 | 0.58 |
| | v1 | | 190 | 0.65 | 0.74 | 0.57 | 0.95 | 0.95 | 0.96 | 0.78 | 0.83 | 0.71 | 0.87 | 0.87 | 0.87 | 0.80 | 0.81 | 0.80 |
| | v2 | | 170 | 0.61 | 0.68 | 0.54 | 0.93 | 0.89 | 0.96 | 0.74 | 0.77 | 0.69 | 0.80 | 0.82 | 0.77 | 0.74 | 0.77 | 0.71 |
| | v3 | | 655 | 0.88 | 0.86 | 0.90 | 0.88 | 0.93 | 0.83 | 0.88 | 0.89 | 0.86 | 0.94 | 0.94 | 0.94 | 0.89 | 0.90 | 0.88 |
| | v4 | | 115 | 0.50 | 0.54 | 0.46 | 1.00 | 1.00 | 1.00 | 0.67 | 0.70 | 0.63 | 0.62 | 0.63 | 0.62 | 0.60 | 0.60 | 0.59 |
| | default | Yes | 1700 | 0.93 | 0.90 | 0.95 | 0.93 | 0.99 | 0.88 | 0.93 | 0.94 | 0.91 | 0.97 | 0.97 | 0.97 | 0.92 | 0.93 | 0.91 |
| | v0 | | 100 | 0.50 | 0.54 | 0.46 | 1.00 | 1.00 | 1.00 | 0.67 | 0.70 | 0.63 | 0.65 | 0.66 | 0.64 | 0.61 | 0.62 | 0.61 |
| | v1 | | 100 | 0.50 | 0.55 | 0.46 | 1.00 | 1.00 | 1.00 | 0.67 | 0.71 | 0.63 | 0.67 | 0.68 | 0.65 | 0.62 | 0.63 | 0.61 |
| | v2 | | 100 | 0.50 | 0.54 | 0.46 | 1.00 | 1.00 | 1.00 | 0.67 | 0.71 | 0.63 | 0.66 | 0.65 | 0.67 | 0.62 | 0.62 | 0.63 |
| | v3 | | 100 | 0.50 | 0.54 | 0.46 | 0.96 | 0.95 | 0.98 | 0.66 | 0.69 | 0.66 | 0.66 | 0.69 | 0.64 | 0.63 | 0.65 | 0.60 |
| | v4 | | 100 | 0.50 | 0.54 | 0.46 | 1.00 | 1.00 | 1.00 | 0.67 | 0.70 | 0.63 | 0.66 | 0.69 | 0.62 | 0.62 | 0.66 | 0.59 |

Test set metrics (image classification rows):

| Category | Architecture | Neg. Ctrl. | Ep. | P a | P l | P h | R a | R l | R h | F1 a | F1 l | F1 h | M1 a | M1 l | M1 h | M2 a | M2 l | M2 h |
|---|---|---|---|---|---|---|---|---|---|---|---|---|---|---|---|---|---|---|
| Image classification | GoogLeNet | No | 42 | 0.89 | 0.92 | 0.85 | 0.86 | 0.78 | 0.95 | 0.87 | 0.84 | 0.90 | 0.90 | 0.93 | 0.87 | 0.85 | 0.88 | 0.81 |
| | Inception v3 | | 70 | 0.83 | 0.80 | 0.86 | 0.84 | 0.82 | 0.85 | 0.84 | 0.81 | 0.86 | 0.57 | 0.61 | 0.53 | 0.53 | 0.58 | 0.48 |
| | EfficientNetB0 | | 10 | 0.86 | 0.83 | 0.90 | 0.87 | 0.87 | 0.87 | 0.86 | 0.85 | 0.88 | 0.76 | 0.79 | 0.74 | 0.72 | 0.75 | 0.69 |
| | Resnext_50 | | 40 | 0.87 | 0.90 | 0.85 | 0.86 | 0.78 | 0.93 | 0.86 | 0.83 | 0.89 | 0.72 | 0.75 | 0.69 | 0.67 | 0.71 | 0.63 |
| | Resnet18 | | 30 | 0.84 | 0.82 | 0.85 | 0.83 | 0.80 | 0.87 | 0.83 | 0.81 | 0.86 | 0.90 | 0.93 | 0.86 | 0.85 | 0.88 | 0.82 |
| | SqueezeNet | | 50 | 0.82 | 0.77 | 0.86 | 0.82 | 0.82 | 0.82 | 0.82 | 0.80 | 0.84 | 0.59 | 0.66 | 0.52 | 0.56 | 0.62 | 0.49 |
| | GoogLeNet | Yes | 26 | 0.84 | 0.88 | 0.80 | 0.81 | 0.68 | 0.93 | 0.82 | 0.77 | 0.86 | 0.92 | 0.94 | 0.91 | 0.88 | 0.90 | 0.86 |
| | Inception v3 | | 42 | 0.85 | 0.85 | 0.84 | 0.84 | 0.77 | 0.90 | 0.84 | 0.81 | 0.87 | 0.59 | 0.63 | 0.55 | 0.55 | 0.59 | 0.50 |
| | EfficientNetB0 | | 15 | 0.86 | 0.86 | 0.87 | 0.86 | 0.82 | 0.90 | 0.86 | 0.84 | 0.89 | 0.59 | 0.65 | 0.53 | 0.55 | 0.61 | 0.49 |
| | Resnext_50 | | 30 | 0.81 | 0.81 | 0.81 | 0.79 | 0.77 | 0.78 | 0.84 | 0.85 | 0.84 | 0.66 | 0.73 | 0.60 | 0.63 | 0.70 | 0.56 |
| | Resnet18 | | 20 | 0.81 | 0.79 | 0.84 | 0.81 | 0.77 | 0.85 | 0.81 | 0.78 | 0.84 | 0.56 | 0.59 | 0.53 | 0.53 | 0.56 | 0.49 |
| | SqueezeNet | | 10 | 0.78 | 0.73 | 0.84 | 0.79 | 0.80 | 0.78 | 0.79 | 0.76 | 0.81 | 0.59 | 0.66 | 0.52 | 0.56 | 0.63 | 0.50 |

**Table 3. Validation and test set metrics of the best neem fruit _D, E, N,_ and _S_ and leaf _A, D, E, N,_ and _S_ models under object detection using _YOLOv5 medium_ variant architecture on a training set augmented with background images.**

| Plant part | Analyte | Ep. | Validation set P a | l | h | R a | l | h | F1 a | l | h | M1 a | l | h | M2 a | l | h | Test set P a | l | h | R a | l | h | F1 a | l | h | M1 a | l | h | M2 a | l | h |
|---|---|---|---|---|---|---|---|---|---|---|---|---|---|---|---|---|---|---|---|---|---|---|---|---|---|---|---|---|---|---|---|---|---|
| Fruit | D | 903 | 0.96 | 0.96 | 0.95 | 0.95 | 0.96 | 0.94 | 0.95 | 0.96 | 0.95 | 0.98 | 0.98 | 0.99 | 0.93 | 0.94 | 0.92 | 0.92 | 0.93 | 0.91 | 0.96 | 0.97 | 0.96 | 0.94 | 0.95 | 0.94 | 0.97 | 0.96 | 0.98 | 0.92 | 0.91 | 0.92 |
| | E | 744 | 0.89 | 0.86 | 0.91 | 0.89 | 0.94 | 0.83 | 0.89 | 0.90 | 0.87 | 0.92 | 0.91 | 0.94 | 0.87 | 0.87 | 0.87 | 0.92 | 0.88 | 0.95 | 0.88 | 0.87 | 0.89 | 0.90 | 0.88 | 0.92 | 0.95 | 0.94 | 0.95 | 0.89 | 0.89 | 0.90 |
| | N | 320 | 0.84 | 0.89 | 0.79 | 0.89 | 0.91 | 0.88 | 0.87 | 0.90 | 0.88 | 0.93 | 0.92 | 0.94 | 0.87 | 0.87 | 0.87 | 0.86 | 0.85 | 0.87 | 0.87 | 0.88 | 0.85 | 0.86 | 0.87 | 0.86 | 0.89 | 0.85 | 0.93 | 0.85 | 0.80 | 0.88 |
| | S | 1500 | 0.97 | 0.96 | 0.98 | 0.97 | 0.98 | 0.96 | 0.97 | 0.97 | 0.97 | 0.99 | 0.99 | 0.99 | 0.94 | 0.95 | 0.93 | 0.97 | 0.98 | 0.96 | 0.97 | 0.96 | 0.98 | 0.97 | 0.97 | 0.97 | 0.98 | 0.97 | 0.98 | 0.93 | 0.92 | 0.93 |
| Leaf | A | 161 | 0.72 | 0.65 | 0.80 | 0.87 | 0.79 | 0.96 | 0.79 | 0.71 | 0.87 | 0.85 | 0.78 | 0.92 | 0.48 | 0.47 | 0.50 | 0.64 | 0.58 | 0.70 | 0.83 | 0.73 | 0.92 | 0.72 | 0.65 | 0.80 | 0.79 | 0.74 | 0.83 | 0.42 | 0.38 | 0.47 |
| | D | 136 | 0.75 | 0.69 | 0.80 | 0.82 | 0.89 | 0.74 | 0.78 | 0.78 | 0.77 | 0.83 | 0.77 | 0.88 | 0.45 | 0.42 | 0.48 | 0.70 | 0.68 | 0.72 | 0.81 | 0.88 | 0.74 | 0.75 | 0.77 | 0.73 | 0.81 | 0.76 | 0.85 | 0.44 | 0.40 | 0.48 |
| | E | 131 | 0.53 | 0.56 | 0.51 | 0.89 | 0.88 | 0.90 | 0.67 | 0.68 | 0.65 | 0.65 | 0.63 | 0.67 | 0.36 | 0.34 | 0.38 | 0.56 | 0.48 | 0.63 | 0.83 | 0.79 | 0.87 | 0.67 | 0.60 | 0.73 | 0.67 | 0.66 | 0.68 | 0.37 | 0.34 | 0.41 |
| | N | 106 | 0.75 | 0.82 | 0.68 | 0.77 | 0.71 | 0.83 | 0.76 | 0.76 | 0.75 | 0.80 | 0.82 | 0.77 | 0.43 | 0.43 | 0.43 | 0.73 | 0.71 | 0.75 | 0.78 | 0.73 | 0.83 | 0.76 | 0.72 | 0.79 | 0.78 | 0.75 | 0.81 | 0.42 | 0.40 | 0.45 |
| | S | 121 | 0.71 | 0.68 | 0.74 | 0.82 | 0.83 | 0.81 | 0.76 | 0.75 | 0.77 | 0.83 | 0.79 | 0.86 | 0.45 | 0.45 | 0.44 | 0.64 | 0.61 | 0.67 | 0.85 | 0.90 | 0.80 | 0.73 | 0.73 | 0.73 | 0.77 | 0.69 | 0.85 | 0.43 | 0.38 | 0.49 |

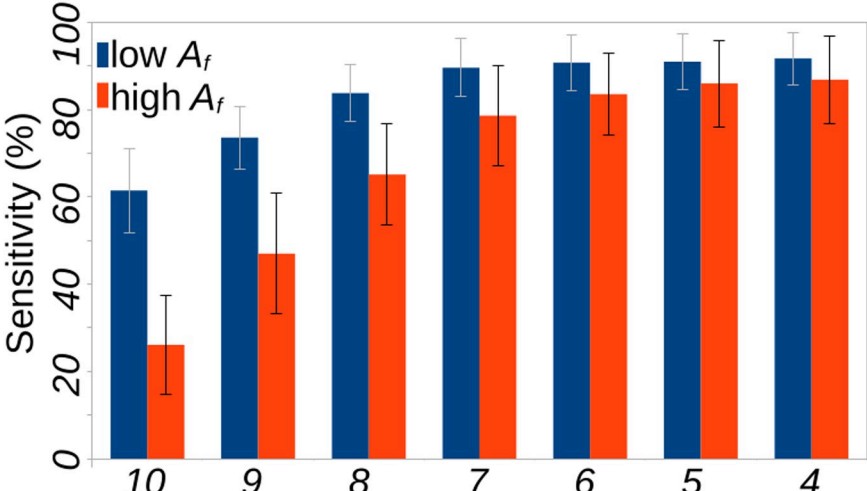

**Fig 4. Boost in fruit *A* prediction sensitivity using multi-metabolite prediction models.** The *X*-and *Y*-axes in the graph indicate the minimum number of dataset shuffles in which the predictive pattern was present and the average sensitivity across those dataset shuffles for the low and high fruit azadirachtin classes respectively. The error bars are the standard deviation (*SD*) in the sensitivity across ten shuffles.

## Discussion

Deep learning-based methods have provided impressive results across several domains, primarily visual and auditory recognition [15]. Data-intensive biological problems are well-suited for deep learning methods [16]. Biologically inspired neural networks are a class of machine learning algorithms that enable learning from data. Deep learning requires a neural network with multiple layers. These methods use supervised, unsupervised, or reinforcement learning-based training models depending on the nature of the data and the type of question asked. Convolutional neural networks (CNNs or ConvNets) are multi-layered neural networks trained with back-propagation algorithms for recognizing images with minimum pre-processing. In 1998, LeCun and co-workers described a machine learning technique that was built to learn automatically with less hand-designed heuristics [17], forming the basis for the development of the CNN field. CNNs combine three architectural ideas: local receptive fields, shared weights, and, sometimes, spatial or temporal sub-sampling [18]. Biological data is often complex, multi-dimensional, and heterogeneous. However, the possibility of using deep learning methods to discover patterns in such large and complex biological datasets is promising. To date, image-based analysis has been used to study plant stress and phenotyping [19–21] and to assess the quality of fruits, grains, and vegetables [22–25].

Computer vision technology has been used previously for fruit and vegetable grading, ripeness detection, quality assessment and evaluation, calorie estimation, disease detection and classification, and sorting in the industry based on one or more of the above parameters [26, 27]. Improved mobile network availability in rural areas facilitates the use of imaging-based agriculture applications without expensive equipment, laboratory procedures, specialized skills, and money. The tools incorporating deep neural networks in horticulture described so far use images and extract features based on color, texture, and shape with known data pre-processing methods, segmentation, feature extraction, classification, and performance measurement. Various colors, shapes, textures, and other feature descriptors make fruit images an ideal input for computer vision. One of the most challenging aspects is collecting and preparing a well-annotated dataset for deep learning applications in agriculture. In our study, we

spent significant time and effort covering a vast geographical area (*0.6* million sq km) to collect fruits and leaves, imaging them, and annotating them using analytical procedures for metabolite concentration measurement in the laboratory. Unlike the methods used so far, which primarily use quality assessment based on different criteria described above, this study links images with the concentration class of an intracellular secondary metabolite.

Our study used fruits from a specific tree (neem) known to bear a high concentration of useful metabolites in its fruits. Although metabolic engineering holds much promise [28], it will take time to use genome and transcriptome sequencing information [29, 30] to understand the pathways and engineer them to mass-produce secondary metabolites. In the short term, selecting trees with high concentrations of metabolites in fruits using simple, easy-to-use, and inexpensive tools will help the industry boost the yield of metabolite production. Considering this, we explored using images alone to predict fruit metabolite concentration class. We collected fruits and leaves from neem trees grown in the wild over a vast area (*0.6* million sq km), imaged them, measured the analytical concentrations of five secondary metabolites in fruits using HPLC and used the fruit and leaf images along with their corresponding metabolite values to test various deep learning-based frameworks. In all the models tested, the *YOLOv5m* default model predicted the best *F1* score (validation: 0.93 and test: 0.88) for fruit azadirachtin. Further, we observed higher sensitivity and specificity under a multi-analyte framework while combining predictions from multi-metabolite models based on azadirachtin, salannin, deacetyl-salannin, nimbin and nimbolide trained from fruit and leaf images compared to results from a single-analyte framework based on only the fruit azadirachtin model. Since each metabolite is linked to specific characteristics, using data from multiple metabolites measured in the same fruit to boost the model's accuracy made sense. This ensures that fruits with high-azadirachtin concentration most likely are enriched in some metabolites while depleted for others. The sensitivity in predicting *low* and *high* classes for fruit *A* was boosted by ~9% and ~6%, respectively. In contrast, the specificity was boosted from 79.40 ± 8.50 and 85.64 ± 6.21, respectively, to 100% for both classes after combining predictions from the single fruit *A* model with that from the other nine models, namely *D*, *E*, *N* and *S* for the fruit, and all five metabolites for the leaf images. Thus, the overall class prediction accuracy was higher with the multi-analyte boosting. Future studies with more multi-dimensional data from the same source may enhance the performance of the multi-analyte framework further.

This study, although simple, represents a significant advancement for the metabolite extraction industry in terms of its utility, being the first to establish a direct link between the image of a fruit or leaf and the concentration class of a secondary metabolite. While our research has practical implications, such as enhancing metabolite production through tree selection, the study has limitations. For instance, we found that the predictive sensitivity of azadirachtin was significantly improved when combined with predictions from nine multi-metabolite models, compared to using the fruit azadirachtin model alone. Although we had far more images (*n = 1045*) than some of the previous studies reported on various fruit sweetness models, the numbers still need to be higher in the context of deep learning. For studies like ours, where images are linked with biological parameters, and unlike popular computer vision problems like facial recognition, it is time-consuming, cumbersome, and expensive to procure a large training dataset. This is especially true where a linked biological metabolite measurement is involved. We did not perform any variance decomposition study. However, multiple factors are likely linked with visible traits of fruits. As a recent study involving tomato and blueberry [31] has shown, our study's neem fruit and leaf images may not reflect agronomic values independently. Additionally, the quality of fruits and leaves vary based on environmental factors and time of collection. Therefore, it is possible that deep learning-based methods that use images may produce a varying accuracy based on the timing of collection of datasets. Future

research with multiple measurements from the same fruits, including acids, metabolites and other compounds, may improve deep learning-based image classification and provide a better underlying predictive model.

## Conclusion

In the first of its kind, the current study extends the application of multiple well-established deep-learning models to neem fruit images to predict the concentration class of a secondary metabolite, azadirachtin. The performance of *YOLOv5m*, the best-performing model, was further boosted under a multi-analyte framework using leaf images and other metabolites. We developed an Android mobile App, *Fruit-In-Sight*, incorporating the multi-analyte framework that uses fruit and leaf images to decide whether to 'pick' or 'not pick' the fruits from a specific tree based on the metabolite concentration class. The tool helps choose suitable fruits for azadirachtin extraction without expensive laboratory equipment and analytical procedures.

## Supporting information

**S1 Table.** Prediction combinations used under the multi-variate framework for low (A) and high (B) fruit azadirachtin class prediction.
(XLSX)

## Acknowledgments

We thank the National Supercomputing Facility PARAM SIDDHI hosted at the Centre for Development of Advanced Computing (CDAC), Pune for providing computing facility for the study.

## Author Contributions

**Conceptualization:** Neeraja M. Krishnan, Binay Panda.

**Data curation:** Neeraja M. Krishnan.

**Formal analysis:** Neeraja M. Krishnan, Saroj Kumar.

**Funding acquisition:** Binay Panda.

**Investigation:** Neeraja M. Krishnan, Binay Panda.

**Methodology:** Neeraja M. Krishnan, Binay Panda.

**Project administration:** Binay Panda.

**Resources:** Binay Panda.

**Software:** Neeraja M. Krishnan.

**Supervision:** Neeraja M. Krishnan, Binay Panda.

**Validation:** Neeraja M. Krishnan, Saroj Kumar.

**Visualization:** Neeraja M. Krishnan.

**Writing – original draft:** Neeraja M. Krishnan, Binay Panda.

**Writing – review & editing:** Neeraja M. Krishnan, Binay Panda.

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
