## [Decision Letter · Decision Letter 0]

30 Apr 2024

PONE-D-24-10515Fruit-In-Sight: a deep learning-based framework for secondary metabolite class prediction using fruit and leaf imagesPLOS ONE

Dear Dr. Panda,

Thank you for submitting your manuscript to PLOS ONE. After careful consideration, we feel that it has merit but does not fully meet PLOS ONE’s publication criteria as it currently stands. Therefore, we invite you to submit a revised version of the manuscript that addresses the points raised during the review process.

We look forward to receiving your revised manuscript.

Kind regards,

Eugenio Llorens

Academic Editor

PLOS ONE

Journal Requirements:

"Research reported in this manuscript is funded by an extramural grant from the Department of Biotechnology, Government of India to BP (BT/PR36744/BID/7/944/2020)."     

Reviewers' comments:

Reviewer's Responses to Questions

**Comments to the Author**

1. Is the manuscript technically sound, and do the data support the conclusions?

Reviewer #1: No

Reviewer #2: Yes

2. Has the statistical analysis been performed appropriately and rigorously? 

Reviewer #1: Yes

Reviewer #2: No

3. Have the authors made all data underlying the findings in their manuscript fully available?

Reviewer #1: Yes

Reviewer #2: Yes

4. Is the manuscript presented in an intelligible fashion and written in standard English?

Reviewer #1: No

Reviewer #2: Yes

5. Review Comments to the Author

Reviewer #1: The presented work has limited novelty as already existing approaches have been applied. It is my advice to introduce innovation and novelty in the approach. Experimental work seems satisfactory

Regards,

Reviewer #2: Authors have identified the best model out of YOLOv5, GoogLeNet,23 InceptionNet, EfficientNet_B0, Resnext_50, Resnet18, and SqueezeNet for detecting metabolite class from fruits and leaf images for plucking in right time. The following points need to be addressed for better understanding by the readers.

The organization need to be improved.

In many location the typo errors are noticed and needs proper care. Some cases the lines are not complete. Its difficult to identify the contributions made in this manuscript.

No conclusion section was avaliable. Discussion need to be improved.

Reference are not presented as per the journal standard. Uniformity in author's name, jnl name, page number, year, vol, and page numbers are missing.

Figures are of not good quality.

6. PLOS authors have the option to publish the peer review history of their article (what does this mean?). If published, this will include your full peer review and any attached files.

Reviewer #1: No

Reviewer #2: **Yes: **Malaya Kumar Nath

---

## [Author Response · Author response to Decision Letter 0]

28 May 2024

Point-to-point rebuttal 

Academic Editor:

Response: We have ensured that the revised manuscript meets the PLOS ONE’s style requirements.

Response: We have deposited the codes in GitHub (https://github.com/binaypanda/Fruit-In-Sight), and added the same under the Data Availability section of the submission.

"Research reported in this manuscript is funded by an extramural grant from the Department of Biotechnology, Government of India to BP (BT/PR36744/BID/7/944/2020)." 

Response: We have added the following disclosure - The funders had no role in study design, data collection and analysis, decision to publish, or preparation of the manuscript.

Response: All data are publicly available at https://www.kaggle.com/datasets/binaypandalabmember/plos-one-data/. We have accordingly modified the Data Availability Statement in the submission form. 

Response: We have included captions to the Supporting Information files at the end of the manuscript and made changes to the file name and in-text citation as per the journal guidelines

Reviewers 1 and 2:

1. Is the manuscript technically sound, and do the data support the conclusions?

Reviewer #1: No

Reviewer #2: Yes

Response: We conducted the analyses rigorously and with appropriate controls. The sample sizes are indicated in Table 1 and the controls in Table 2. With the same seed, we ensure that the runs are reproducible. Furthermore, we have provided all the codes used in the study on Github (https://github.com/binaypanda/Fruit-In-Sight).

2. Has the statistical analysis been performed appropriately and rigorously?

Reviewer #1: Yes

Reviewer #2: No

Response: To ensure the predicted model's rigour, we performed bootstrapping using 10-fold cross-validation for both the single-analyte and multi-analyte frameworks in addition to the original run to avoid any training bias. We report these cross-validation errors (line # 247-248).

3. Have the authors made all data underlying the findings in their manuscript fully available?

Reviewer #1: Yes

Reviewer #2: Yes

Response: Thank you.

4. Is the manuscript presented in an intelligible fashion and written in standard English?

Reviewer #1: No

Reviewer #2: Yes

Response: We have gone over the manuscript carefully and removed typographical and grammatical errors in the revised manuscript.

Reviewer 1: 

Comment: The presented work has limited novelty as already existing approaches have been applied. It is my advice to introduce innovation and novelty in the approach. 

Response: We appreciate the reviewer’s comments. As a biology lab, we leverage and optimise existing methods for novel applications. Applications of existing algorithms and tools are often crucial in areas where they were not originally intended for. The current study, we believe, is a testament to that approach. One of the most challenging aspects of our research was collecting and preparing a well-annotated dataset for deep-learning applications in agriculture. We dedicated a significant amount of time and effort to cover a vast geographical area (0.6 million sq. km), collecting fruits and leaves, imaging them, and annotating them before using them for the analytical metabolite concentration procedure in the laboratory. As the reviewers will appreciate, this was not without challenges, often in some places with summer temperatures (the tree only fruits during the summer months) of close to 50 degrees C. 

The other aspect of our work is a thorough comparison with multiple available tools. Other than the default YOLOv5m architecture, we tested five variants: v0 (the YOLO architecture adapted to detect small objects (http://cs230.stanford.edu/projects_fall_2021/reports/103120671.pdf)), v1 (modi-fied head by tweaking v0 and increasing the number of C3 layers in the p2 block from 1 to 3, in the p3 block from 3 to 5, in the p4 block from 3 to 5 and added a p5 block), v2 (added an extra p2 block to head), v3 (added 1 extra p2 and p5 blocks each to the head), and v4 (deleted p5). Using default param-eters, we also tested six state-of-the-art architectures under image classification frameworks such as GoogLeNet, Inception v3, EfficientNet_B0, Resnext_50, Resnet18 and SqueezeNet. The image classifi-cation frameworks used cropped images after bounding box detection using the best model obtained from YOLOv5.

For the best model, we find compelling evidence that boosting a single-analyte model predict-ing azadirachtin, with predictions from nine other image-analyte models, significantly increases the azadirachtin prediction sensitivity and results in complete specificity. To our knowledge, this combina-torial approach is a novel method to enhance the prediction accuracy of a model while utilizing multi-dimensional outputs from the same specimen using respective models. Furthermore, we have devel-oped a mobile application that leverages this power to predict neem azadirachtin class in real-time on the field.

The above efforts add innovation and novelty towards a new application to the prevailing ob-ject detection and classification paradigms. 

Comment: Experimental work seems satisfactory

Response: Thank you.

Reviewer 2:

Authors have identified the best model out of YOLOv5, GoogLeNet,23 InceptionNet, EfficientNet_B0, Resnext_50, Resnet18, and SqueezeNet for detecting metabolite class from fruits and leaf images for plucking in right time. The following points need to be addressed for better understanding by the readers.

Comment: The organization need to be improved.

Response: We thank the reviewer for this comment. We have gone over the manuscript, organized it better and improved the flow and clarity. All the modified texts are highlighted in the revised submission.

Comment: In many location the typo errors are noticed and needs proper care. Some cases the lines are not complete. 

Response: Thank you. We have gone over the manuscript and removed the typographical and grammar errors. 

Comment: It’s difficult to identify the contributions made in this manuscript. No conclusion section was available. 

Response: We thank the reviewer for pointing this out. Following the suggestion, we have added a separate Conclusion section (line # 368-375) and highlighted the findings and contributions made in the manuscript. 

Comment: Discussion need to be improved.

Response: We have re-organized the Discussion section, added additional content (line # 304-319 and line # 346-352) and improved the flow and clarity. The revised portions are highlighted in the text. 

Comment: Reference are not presented as per the journal standard. Uniformity in author's name, jnl name, page number, year, vol, and page numbers are missing.

Response: We have formatted the references as per the journal standard.

Comment: Figures are of not good quality.

Response: We have uploaded higher quality images.

---

## [Decision Letter · Decision Letter 1]

25 Jun 2024

PONE-D-24-10515R1Fruit-In-Sight: a deep learning-based framework for secondary metabolite class prediction using fruit and leaf imagesPLOS ONE

Dear Dr. Panda,

Thank you for submitting your manuscript to PLOS ONE. After careful consideration, we feel that it has merit but does not fully meet PLOS ONE’s publication criteria as it currently stands. Therefore, we invite you to submit a revised version of the manuscript that addresses the points raised during the review process.

We look forward to receiving your revised manuscript.

Kind regards,

Eugenio Llorens

Academic Editor

PLOS ONE

Reviewers' comments:

Reviewer's Responses to Questions

**Comments to the Author**

1. If the authors have adequately addressed your comments raised in a previous round of review and you feel that this manuscript is now acceptable for publication, you may indicate that here to bypass the “Comments to the Author” section, enter your conflict of interest statement in the “Confidential to Editor” section, and submit your "Accept" recommendation.

Reviewer #1: (No Response)

2. Is the manuscript technically sound, and do the data support the conclusions?

Reviewer #1: Partly

3. Has the statistical analysis been performed appropriately and rigorously? 

Reviewer #1: I Don't Know

4. Have the authors made all data underlying the findings in their manuscript fully available?

Reviewer #1: Yes

5. Is the manuscript presented in an intelligible fashion and written in standard English?

Reviewer #1: Yes

6. Review Comments to the Author

Reviewer #1: Still I am not convinced with the experimentation process and regarding novelty and scientific contribution

Please look into it and resolve the issue

7. PLOS authors have the option to publish the peer review history of their article (what does this mean?). If published, this will include your full peer review and any attached files.

Reviewer #1: No

---

## [Author Response · Author response to Decision Letter 1]

26 Jul 2024

Point-to-point rebuttal 

Reviewer's Responses to Questions

Comments to the Author

1. If the authors have adequately addressed your comments raised in a previous round of review and you feel that this manuscript is now acceptable for publication, you may indicate that here to bypass the “Comments to the Author” section, enter your conflict of interest statement in the “Confidential to Editor” section, and submit your "Accept" recommendation.

Reviewer #1: (No Response)

2. Is the manuscript technically sound, and do the data support the conclusions?

Reviewer #1: Partly

Authors’ response: We conducted the analyses rigorously with appropriate controls and with statistical rigor. Additionally, we have described the methods in detail, made all the data and code openly available in Kaggle (https://www.kaggle.com/datasets/binaypandalabmember/plos-one-data/ ) and Github ((https://github.com/binaypanda/Fruit-In-Sight), and have given all the background information for anyone to reproduce our results. Therefore, we believe that our manuscript presents a technically sound piece of scientific research.

3. Has the statistical analysis been performed appropriately and rigorously? 

Reviewer #1: I Don't Know

Authors’ response: As mentioned above, we have conducted the analyses rigorously and with appropriate controls and with statistical rigor, where required.

4. Have the authors made all data underlying the findings in their manuscript fully available?

Reviewer #1: Yes

5. Is the manuscript presented in an intelligible fashion and written in standard English?

Reviewer #1: Yes

6. Review Comments to the Author

Reviewer #1: Still I am not convinced with the experimentation process and regarding novelty and scientific contribution.

Please look into it and resolve the issue.

Authors’ response: As scientists, we regularly review manuscripts from many journals written by other scientists, and therefore, we respect the reviewer's comments. However, we disagree with the reviewer's comment on "novelty and scientific contribution". As mentioned before, innovations in biology do not only sometimes happen due to the development of new algorithms but also because of how ingenious we are in using the tools to develop new applications. Additionally, the collection of new, good-quality and well-annotated data is something that we must emphasize. The quality of science is often dependent on this. Our manuscript describes a new application of deep-learning tools using a dataset that took years to collect, annotate and analyze. Therefore, when we hear that the reviewer is not convinced, we have to know the details of the points the reviewer is not convinced of. 

The revised manuscript and our response to the reviewer's comments earlier on the same point adequately addressed this point. For clarity, we would like to repeat that again below - 

As a biology lab, we leverage and optimise existing methods for novel applications. Applications of existing algorithms and tools are often crucial in areas where they were not originally intended for. The current study, we believe, is a testament to that approach. One of the most challenging aspects of our research was collecting and preparing a well-annotated dataset for deep-learning applications in agriculture. We dedicated a significant amount of time and effort to cover a vast geographical area (0.6 million sq. km), collecting fruits and leaves, imaging them, and annotating them before using them for the analytical metabolite concentration procedure in the laboratory. As the reviewers will appreciate, this was not without challenges, often in some places with summer temperatures (the tree only fruits during the summer months) of close to 50 degrees C. 

The other aspect of our work is a thorough comparison with multiple available tools. Other than the default YOLOv5m architecture, we tested five variants: v0 (the YOLO architecture adapted to detect small objects (http://cs230.stanford.edu/projects_fall_2021/reports/103120671.pdf)), v1 (modified head by tweaking v0 and increasing the number of C3 layers in the p2 block from 1 to 3, in the p3 block from 3 to 5, in the p4 block from 3 to 5 and added a p5 block), v2 (added an extra p2 block to head), v3 (added 1 extra p2 and p5 blocks each to the head), and v4 (deleted p5). Using default parameters, we also tested six state-of-the-art architectures under image classification frameworks such as GoogLeNet, Inception v3, EfficientNet_B0, Resnext_50, Resnet18 and SqueezeNet. The image classification frameworks used cropped images after bounding box detection using the best model obtained from YOLOv5.

For the best model, we find compelling evidence that boosting a single-analyte model predicting azadirachtin, with predictions from nine other image-analyte models, significantly increases the azadirachtin prediction sensitivity and results in complete specificity. To our knowledge, this combinatorial approach is a novel method to enhance the prediction accuracy of a model while utilizing multi-dimensional outputs from the same specimen using respective models. Furthermore, we have developed a mobile application that leverages this power to predict neem azadirachtin class in real-time on the field.

The above efforts add innovation and novelty towards a new application to the prevailing object detection and classification paradigms. 

7. PLOS authors have the option to publish the peer review history of their article (what does this mean?). If published, this will include your full peer review and any attached files.

Do you want your identity to be public for this peer review? For information about this choice, including consent withdrawal, please see our Privacy Policy.

Reviewer #1: No

---

## [Editor Report · Decision Letter 2]

30 Jul 2024

Fruit-In-Sight: a deep learning-based framework for secondary metabolite class prediction using fruit and leaf images

PONE-D-24-10515R2

Dear Dr. Panda,

We’re pleased to inform you that your manuscript has been judged scientifically suitable for publication and will be formally accepted for publication once it meets all outstanding technical requirements.

Kind regards,

Eugenio Llorens

Academic Editor

PLOS ONE
---

## [Editor Report · Acceptance letter]

31 Jul 2024

PONE-D-24-10515R2 

PLOS ONE

Dear Dr. Panda, 

I'm pleased to inform you that your manuscript has been deemed suitable for publication in PLOS ONE. Congratulations! Your manuscript is now being handed over to our production team.

Kind regards, 

on behalf of

Dr. Eugenio Llorens 

Academic Editor

PLOS ONE